# Fibroblast Growth Factor 23 is Associated with a Frequent Exacerbator Phenotype in COPD: A Cross-Sectional Pilot Study

**DOI:** 10.3390/ijms20092292

**Published:** 2019-05-09

**Authors:** Swati Gulati, J. Michael Wells, Gisel P. Urdaneta, Kira Balestrini, Isabel Vital, Katherine Tovar, Jarrod W. Barnes, Surya P. Bhatt, Michael Campos, Stefanie Krick

**Affiliations:** 1Division of Pulmonary, Allergy and Critical Care Medicine, Department of Medicine, The University of Alabama at Birmingham, Birmingham, AL 35294, USA; sgulati@uabmc.edu (S.G.); jmwells@uabmc.edu (J.M.W.); jbarnes@uabmc.edu (J.W.B.); sbhatt@uabmc.edu (S.P.B.); 2UAB Lung Health Center, Birmingham, AL 35294, USA; 3Birmingham VA Medical Center, Birmingham, AL 35294, USA; 4Pulmonary Section, Miami VA Medical Center, Miami, FL 33125, USA; Gisel.UrdanetaCestary@va.gov (G.P.U.); kirabalestrini@gmail.com (K.B.); isabel.vital@va.gov (I.V.); kvtsanchez@gmail.com (K.T.); 5Division of Pulmonary, Allergy, Critical Care and Sleep Medicine, Department of Medicine, University of Miami Leonard M. Miller School of Medicine, Miami, FL 33136, USA

**Keywords:** FGF23, COPD, inflammation

## Abstract

Chronic Obstructive Pulmonary Disease (COPD) is a chronic inflammatory airway disease punctuated by exacerbations (AECOPD). Subjects with frequent AECOPD, defined by having at least two exacerbations per year, experience accelerated loss of lung function, deterioration in quality of life and increase in mortality. Fibroblast growth factor (FGF)23, a hormone associated with systemic inflammation and altered metabolism is elevated in COPD. However, associations between FGF23 and AECOPD are unknown. In this cross-sectional study, individuals with COPD were enrolled between June 2016 and December 2016. Plasma samples were analyzed for intact FGF23 levels. Logistic regression analyses were used to measure associations between clinical variables, FGF23, and the frequent exacerbator phenotype. Our results showed that FGF23 levels were higher in frequent exacerbators as compared to patients without frequent exacerbations. FGF23 was also independently associated with frequent exacerbations (OR 1.02; 95%CI 1.004–1.04; *p* = 0.017), after adjusting for age, lung function, smoking, and oxygen use. In summary, FGF23 was associated with the frequent exacerbator phenotype and correlated with number of exacerbations recorded retrospectively and prospectively. Further studies are needed to explore the role of FGF 23 as a possible biomarker for AECOPD to better understand the pathobiology of COPD and to help develop therapeutic targets.

## 1. Introduction

Chronic obstructive pulmonary disease (COPD) is a progressive disease characterized by chronic airway inflammation and progressive decline in lung function which currently accounts as the third leading cause of death in the United States [1,2]. The natural history of COPD is modified by the presence of acute exacerbations of COPD (AECOPD), which significantly increase airway inflammation and are associated with a more accelerated decline in lung function, negatively impacting quality of life and increasing mortality [3,4,5,6]. Given the variable occurrence of AECOPD among subjects with COPD, recent reports have demonstrated the existence and clinical relevance of a frequent exacerbator subgroup of patients with COPD [7]. Subjects with this particular phenotype do have more airflow limitation, symptoms and health-related quality of life impairment and respiratory disability [7,8,9,10]. Frequent exacerbators demonstrate persistent systemic and airway inflammation thereby entering a destructive cycle of increased cardiovascular risk, worsened comorbidities, frequent hospitalization and mortality [11,12,13]. So far, specific therapeutic approaches targeting this phenotype are sparse [14,15]. Furthermore, it is still not clear whether the COPD frequent exacerbator group has a stable phenotype or whether the rate of exacerbations can change over time [16]. Therefore, it is important to better understand the underlying pathophysiology of this phenotype and identify reliable prognostic biomarkers and novel targets for therapy.

Fibroblast Growth Factor (FGF)23 is a circulating pro-inflammatory hormone and may have a role in affecting lung inflammation in COPD. Circulating FGF23 has been shown to be elevated in COPD patients as compared to age-matched controls [17]. In chronic kidney disease (CKD), FGF23 is a well characterized prognostic marker not only for progression of CKD but also for increased systemic inflammation and mortality in these patients [18,19,20]. It has been shown that circulating FGF23 levels depend on phosphate intake and fractional excretion of phosphate and therefore differ between populations due to differences in diet [21]. Therefore, there is not a universal “normal value”, but previous studies, including ours, have shown that normal values range between 30−40 pg/mL in cohorts similarly to ours [22,23]. 

To our knowledge, no studies have evaluated associations between FGF23 and AECOPD, including the frequent exacerbator phenotype. We hypothesized that given its association with COPD status and systemic inflammation, FGF23 would be elevated among individuals with frequent exacerbations and would be independently associated with the frequent exacerbator phenotype.

## 2. Results

### 2.1. Cohort Characteristics

We enrolled 70 participants in the study. Participants were on average 69 ± 6 years old with mean FEV_1_ 46.5 ± 15.5% predicted. Baseline characteristics of the study population are shown in Table 1. The exacerbations recorded in our cohort where similar for the years before and after study enrollment. Retrospectively, our study cohort had a mean AECOPD frequency of 0.67 ± 1.22 (median 0, IQR 0, 1) for the year before enrollment, with only 35.7% of subjects experiencing at least one exacerbation and 28.5% experiencing at least one moderate or severe exacerbation. For the year following enrollment and FGF23 measurement, the exacerbation rate was 0.64 ± 1.09 (median 0, IQR 0, 1) with 34.2% experiencing at least one exacerbation and 31.4% at least one moderate to severe exacerbation. We considered frequent exacerbators subjects with 2 or more moderate or severe events in the 12 months [7,24]. A total of 11.4% of our population had this phenotype based on past events and 14.2% based on future events. Frequent exacerbators, defined based on events before FGF23 measurement, were not significantly different compared to the infrequent exacerbator group in several demographic and clinical features of COPD (Table 2).

### 2.2. FGF23 and Past COPD Exacerbations

Our study population (all 70 patients) had mean plasma FGF23 levels of 56.5 ± 35.3 pg/mL. FGF23 levels were lower among individuals without any exacerbation in the previous year (49.4 ± 25.6 pg/mL) compared to those who had 1 (53.0 ± 24.9 pg/mL) or ≥2 events (79.5 ± 56.6 pg/mL, *p* = 0.02 by 1-way ANOVA) (Figure 1). FGF23 was elevated in frequent exacerbators as compared to patients with one or no exacerbation (92.7 ± 61.3 vs 51.8 ± 27.9 pg/mL, *p* = 0.001).

In univariate logistic regression models, age, pack year history of smoking, supplemental oxygen use, and FGF23 were associated with the frequent exacerbator phenotype. In a multivariable logistic regression model adjusting for these variables in addition to FEV_1_ percent predicted, FGF23 (OR 1.02; 95%CI 1.004–1.04; *p* = 0.019) was independently associated with the frequent exacerbator phenotype as shown in Table 3.

Furthermore, FGF23 is also independently associated with supplemental oxygen use (OR 4.85; 95%Cl 1.18–19.9; *p* = 0.028).

### 2.3. FGF23 and Future COPD Exacerbations

Frequent exacerbators, now defined based on moderate to severe events that occurred 12 calendar months after the plasma collection, also had higher levels of baseline FGF23 levels (81.1 ± 55.1 pg/mL compared with 52.4 ± 29.4 pg/mL among infrequent exacerbators, *p* = 0.016). The magnitude of these differences in FGF23 levels noted between frequent and infrequent exacerbators was similar to what was noted in the retrospective analysis (Figure 2). FGF23 levels had a positive and significant correlation with the number of total exacerbations during the year following enrollment (R^2^ 0.13, *p* = 0.001), in particular the number of moderate or severe exacerbations (R^2^ 0.16, *p* = 0.0005) as shown in Figure 2. Although not statistically significant, we observed a positive trend in the proportion of subjects that became frequent exacerbators when FGF23 levels were divided in quartiles. While only 5.5% of frequent exacerbators occurred in the lowest FGF23 quartile (<36.2 pg/mL), this proportion rose to 11.7%, 16.6% and 23.5% for quartile 2 (36.2–46.7 pg/mL), quartile 3 (46.8–63.6 pg/mL) and quartile 4 (>63.6 pg/mL) respectively. It has been described that the frequent exacerbator phenotype does exhibit stability over time (16). In our cohort, 38% of frequent exacerbators in the year before sampling also were considered frequent exacerbators in the year after. These subjects had considerably high FGF23 levels (127.3 ± 18.5 pg/mL).

We did not screen for cardiovascular disease as potential confounder, but 10% of our patient cohort were taking angiotensin converting enzyme inhibitors, 22% of them were taking beta blockers (these patients had similar FGF23 distributions as the COPD patient without cardiovascular medication), and 14% were taking both, suggestive of some degree of cardiovascular disease. When comparing plasma FGF23 levels between these groups, we did not see any significant difference of FGF23 levels by ANOVA (Figure 3), suggesting that FGF23 elevations are a primary consequence of COPD. We did not screen for hepatic, endocrine or autoimmune disorders, which could also influence FGF23 levels [25,26].

## 3. Discussion

Among patients with stable COPD, FGF23 was associated with a “frequent exacerbator” phenotype and the frequency of exacerbations in both our retrospective (exacerbations before plasma sampling) and prospective (after plasma sampling) analysis. These findings provide insight into potential mechanisms that contribute to frequent COPD exacerbations.

Multiple studies have shown a strong association between increased FGF23 levels and risk of progression of chronic kidney disease, cardiovascular events and mortality [18,27]. Furthermore, FGF23 levels have been shown to be elevated in various acute and chronic inflammatory diseases [23,28,29,30,31]. However, the role of FGF23 in COPD pathophysiology is not well studied. We have shown in a previous study that individuals with mild-to-moderate COPD, who also had increased systemic and airway inflammation, showed increased FGF23 plasma levels [23]. The cohort in this study did not show any differences in markers of systemic inflammation and we did not see any significant differences in FGF23 levels except the described correlation with COPD exacerbation, which implies FGF23 being a pro-inflammatory marker for COPD. A different study observed higher FGF23 levels in COPD patients with hypophosphatemia as compared to COPD patients with normal phosphate levels [17]. This is interesting as hypophosphatemia in COPD is usually associated with diminished muscle strength at baseline [32,33]. Muscle strength is further decreased during an acute exacerbation and frequent exacerbators often experience worsening muscle weakness with “muscle disuse” from systemic inflammation and reduction in physical activity [34,35,36]. In our cohort, we did not collect information regarding phosphate or respiratory muscle strength at the time of enrollment and this hypothesis needs to be further explored.

Furthermore, it is very interesting that there is an independent association of FGF23 with supplemental oxygen use pointing towards an association between hypoxemia and FGF23. Future studies are needed to further evaluate this association in vitro and in vivo models.

Another unexplored pathway is the relation of FGF23 and lung inflammation. It is well known that frequent exacerbators experience worse outcomes related to increased airway inflammation. Our findings of increased levels of this pro-inflammatory hormone in this subpopulation of COPD subjects, associated with both retrospective and prospective events, points towards as potential role of FGF23 in lung inflammation that needs to be further explored. Recent reports have shown that FGF23 can induce inflammation by directly targeting hepatocytes to release C-reactive protein and by increasing interleukin-8 in the bronchial epithelium [37,38]. In addition, its inverse association with klotho, an anti-aging protein [39,40], may also highlight another avenue that needs to be further explored in COPD.

Our study has several limitations. It is a small cross-sectional study with relatively short follow up time and in stable and mainly male and obese COPD patients, reflecting a veteran population. In our study, we excluded patients with chronic kidney disease (elevated serum creatinine levels). Furthermore, the frequent exacerbator group was small. As mentioned above, our findings will need to be correlated with phosphate levels, measures of inspiratory and expiratory muscle strength and markers of systemic and lung inflammation. In addition, it will be very interesting to measure serial FGF23 levels in COPD cohorts to obtain longitudinal FGF23 trajectories and correlate these with AECOPD frequency and mortality, since it has been shown that a subgroup of CKD patients with rising FGF23 levels has an exceptionally high risk of death [20]. The strengths of our study include the novelty of the findings, with significant associations with exacerbations despite the low number of events in both retrospective and prospective analyses. The strength of the association was independent of other conditions associated with high FGF23 such as cardiovascular disease or renal disease, which were the same between groups.

## 4. Materials and Methods

### 4.1. Study Population

The cross-sectional pilot study was conducted at the Veterans Administration Hospital Miami from June 2016 to October 2016. The Miami VA COPD Cohort, is a longitudinal study of the natural history of veterans with COPD. Subjects were recruited during their stable state in clinics and lung function laboratory and followed prospectively over time. Data collection included a comprehensive documentation of demographic features, including smoking and environmental exposures, exacerbation history and treatments. Questionnaires and chart review were applied to document comorbidities, quality of life and prognostic markers. Other data collected included radiologic data (computed tomography (CT) scans) and complete physiologic data obtained following American Thoracic Society standards. Plasma was sampled at several time points to study biomarkers. Inclusion criteria to participate was to have a diagnosis of COPD confirmed by a pulmonary specialist (based on the presence of airflow obstruction assessed by spirometry, including forced expiratory volume in 1 second (FEV1), forced vital capacity (FVC) [41], symptoms and exposure) and willing to sign the inform consent to participate. The study protocol was approved by the Human Studies Subcommittee of the Institutional Review Board of the Miami VA Healthcare System on 11 May 2016 (ID: 01265) and written consent was obtained from each patient enrolled in the study. For this particular study, we analyzed a cross-section of participants while in their stable state and not treated for a suspected AECOPD within the previous 30 days. Pulmonary function tests performed within the last year during clinical stability were included in the analysis.

### 4.2. Ascertainment of COPD Exacerbations

A COPD exacerbation was defined as a sustained worsening beyond the normal day-to-day variation of the individual’s condition from a stable state that was acute in onset, lasted more than 48 h, and warranted additional treatment [42,43]. During the index visit, participants were asked to self-report the number of times they experienced a COPD exacerbation (any event including hospitalization) over the previous 12-months. Additionally, they were given a questionnaire to gather information about frequency of COPD exacerbations requiring hospitalizations or emergency room visits in the past year. Exacerbations were recorded as count events for analysis. Careful chart review was performed to complement the data recorded as most veterans receive their entire care in the same institution. For the prospective assessment of exacerbations, we had a similar approach, interviewing subjects in their 3-month follow-ups and reviewing charts to document emergency room (ER) or admissions because of AECOPD. We classified as mild those AECOPD treated in clinic, moderate if they warrant an ER visit, and severe if they warranted hospitalization [24].

### 4.3. Blood Sampling and FGF23 Measurement

Ten (10) mL of venous blood was sampled during the index visit. The blood was processed immediately by centrifugation and the plasma fraction was collected and frozen at −80 °C for subsequent processing. Specimen were thawed within 6 months and FGF23 was measured by enzyme-linked immunosorbent assay (ELISA; Immutopics; Athens, OH, USA) as previously described [18,19,38].

### 4.4. Statistical Analysis

Data were expressed as mean ± standard deviation (SD). Participants were defined as “frequent exacerbators”, if they experienced 2 or more moderate or severe exacerbations in the 12 months prior to study enrollment [24], and those with one or no exacerbation were grouped as “infrequent exacerbators”. Subjects were then re-classified based on exacerbations that occurred in the 2 months after enrollment. Bivariate analyses included unpaired *t*-testing for measuring between group differences for continuous variables or chi-square testing or Fisher’s exact testing for categorical variables, as appropriate. One-way ANOVA was used to measure differences in FGF23 between individuals with 0, 1, and ≥2 exacerbations in the prior year. Logistic regression analysis was conducted to determine associations between clinical data, lung function, assessed by spirometry [41], FGF23, and frequent exacerbators. Variables significantly associated with the frequent exacerbator phenotype on univariate logistic regression models with a *p*-value < 0.05 were entered in the final multivariable logistic regression model plus FEV1 percent predicted based on well-established associations with frequent exacerbations in the literature [6,13,43]. We used SPSS (version 24.0, Chicago, IL, USA) for all statistical analyses. Statistical significance was set at *p*-value < 0.05. 

## 5. Conclusions

FGF23 was independently associated with the frequent exacerbator phenotype of COPD both retrospectively and prospectively. These results provide evidence that the FGF23 pathway needs to be further studied in the context of AECOPD as it may serve as a therapeutic area for future studies as a potential biomarker and pro-inflammatory mediator, as previously shown [23].

## Figures and Tables

**Figure 1 ijms-20-02292-f001:**
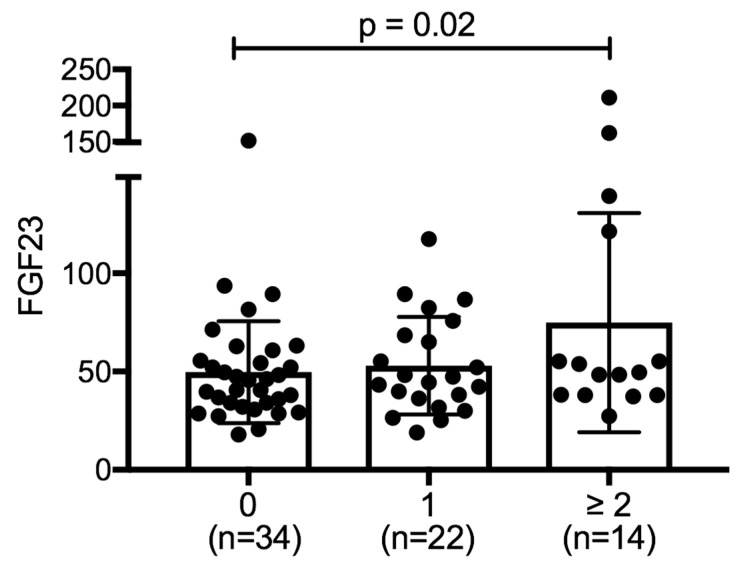
Fibroblast Growth Factor (FGF) 23 levels across chronic obstructive pulmonary disease (COPD) exacerbation frequency sub-groups. Box and dot plot indicating FGF23 plasma level means in COPD patients sub-grouped by exacerbations with 0 = no exacerbations, 1 = one exacerbation and >2 exacerbations during the study period (statistical analysis using 1 way ANOVA with *p* = 0.02).

**Figure 2 ijms-20-02292-f002:**
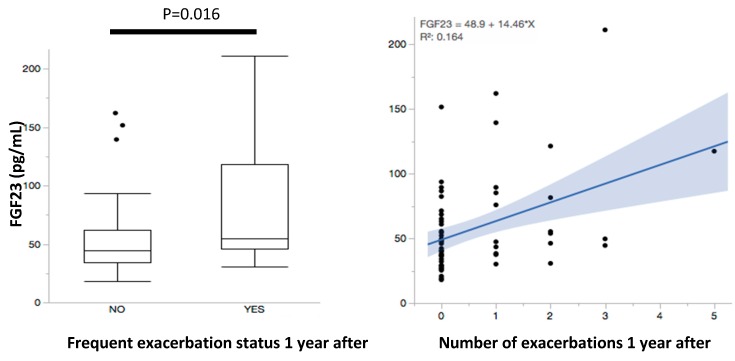
FGF23 levels and correlation with number of exacerbations 12 months after FGF23 sampling. The left panel shows a whisker plot for FGF23 plasma levels in individuals without and with frequent exacerbations that occurred 12 months after sampling for FGF23 measurements. The right panel illustrates the correlation (shaded band representing the 95% confidence interval of the fitted values) between FGF23 plasma levels and number of exacerbations for this time frame.

**Figure 3 ijms-20-02292-f003:**
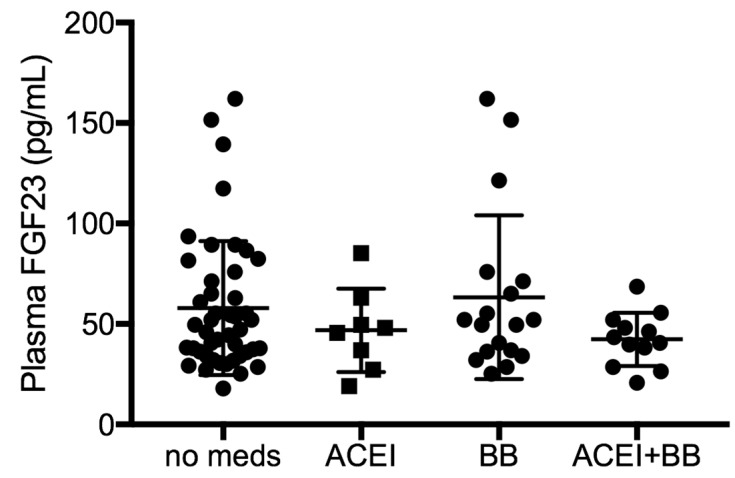
FGF23 levels in individuals with COPD, subgrouped according to their use of cardiovascular medication. Dot plot showing plasma FGF23 levels in individuals with COPD, who receive no cardiac medications (no meds), or regularly take ACE inhibitors (ACEI), beta blockers (BB) or both (ACEI + BB), showing no significant differences in FGF23 levels using multiple comparison ANOVA.

**Table 1 ijms-20-02292-t001:** Baseline characteristics of study population.

Variable (*n* = 70)	
Age, years (SD)	69 (6)
Male, %	94%
White race, %	69%
BMI, kg/m^2^ (SD)	25.6 (5.5)
FEV_1_, %predicted (SD)	46.5 (15.5)
FEV_1_/FVC, (SD)	0.44 (0.11)
Current smokers, %	26%
Smoking pack years, (SD)	60 (37)
Supplemental oxygen use	18 (26%)
Serum WBC (10^3^ cells/mm^3^) (SD)	8.5 (7.1)
Plasma FGF23, pg/mL, (SD)	56.5 (35.3)
Serum Creatinine (mg/dL) (SD)	1.0 (0.2)

**Table 2 ijms-20-02292-t002:** Baseline characteristics in groups based on exacerbation frequency.

Variable, *n* (SD)	Infrequent Exacerbators (*n* = 62) *	Frequent Exacerbators (*n* = 8) ^#^	*p* Value
Age, years (SD)	68.4 (5.7)	70.2 (4.8)	0.386
Male, %	58 (93.5%)	8 (100%)	0.323
White race	47 (75.8%)	3 (37.5%)	0.037
BMI, mg/k^2^ (SD)	25.6 (5.8)	26.2 (3.4)	0.769
FEV1, % (SD)	46.5 (15.6)	46.5 (16.0)	1.00
FEV1/FVC,	43.7 (11.4)	47.7 (12.3)	0.359
Creatinine, mg/dL (SD)	0.99 (0.22)	1.0 (0.29)	0.296
Current smoker	17 (27.4%)	1 (12.5%)	0.539
Smoking pack years	60.7 (38.6)	55.2 (19.3)	0.696
Supplemental O_2_ use	15 (24.2%)	3 (37.5%)	0.412
CAT score	17.6 (9.0)	16.2 (7.7)	0.712
mMRC dyspnea scale	1.91 (1.3)	1.85 (1.3)	0.906
SGRQ, Total	46.7 (21.4)	50.0 (26.2)	0.748
Charlson Index	5.0 (2.1)	6.8 (5.0)	0.068
BODE score	4.5 (2.0)	4.1 (2.0)	0.670

* Non frequent exacerbators are defined as patients with less than 2 or no exacerbations in 12 months prior to enrollment; ^#^ Frequent exacerbators are defined as patients with 2 or more moderate and severe exacerbations in 12 months prior to enrollment.

**Table 3 ijms-20-02292-t003:** Associations with the frequent exacerbator phenotype.

Variable	Unadjusted Model			Adjusted Model		
	OR	95% CI	*p* Value	OR	95% CI	*p* Value
Age	1.13	1.00–1.26	0.043	1.04	0.90–1.21	0.601
Smoking, pack years	0.97	0.95–1.00	0.046	0.98	0.95–1.01	0.106
Post-BD FEV1, %	0.99	0.95−1.03	0.683			
WBC	1.10	0.94–1.29	0.228			
Creatinine	2.34	0.22–25.1	0.483			
FGF23	1.02	1.00–1.04	0.017	1.02	1.004–1.04	0.019
Suppl. Oxygen Use	4.09	1.19–14.1	0.026	4.85	1.18–19.9	0.028

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
