# Peer review of "Fibroblast Growth Factor 23 is Associated with a Frequent Exacerbator Phenotype in COPD: A Cross-Sectional Pilot Study"

_ijms, 2019, doi:10.3390/ijms20092292_

Round 1
Reviewer 1 Report
FGF23 plays an essential role in maintaining serum phosphate homeostasis. It has been shown that plasma concentrations of FGF23 are elevated and strongly associated with the risk of chronic kidney disease CKD progression, cardiovascular events and mortality. FGF23 is also elevated in Chronic Obstructive Pulmonary Disease (COPD). In this manuscript, the authors investigated the association between plasma FGF23 levels and AECOPD (Acute exacerbation of COPD). They found that FGF23 levels were higher in frequent exacerbations. Although the sample number is very small and the demographic is an almost exclusively male patients, the results sound significant and could support the possibility that plasma FGF23 levels may serve as a potential proinflammatory mediator.
The reviewer has a few questions and some comments.
1. According to the reference (PMID: 29748308), Individuals with mild-to-moderate COPD had higher FGF23 plasma levels when compared to patients without COPD (59±12 versus 42±10 pg/ml, P=0.004; Fig. 1a). However, FGF23 was not significantly increased in severe COPD compared to controls in this study population (51 ± 18 versus 42±10 pg/ml, P=0.26). What is the difference between severe COPD and AECOPD? This is not very clear and needs some clarification.
2. Are there any available to look at direct associations between FGF23 levels and COPD exacerbations in individuals who had the exacerbation events only after the initiation of the study? This study could have been better and much more informative, if they measured FGF levels both at the beginning and end of the study.
3. How does individual FGF23 levels fluctuate with time?
4. In the method section, the reviewer found the statement of “Pulmonary function tests performed … were included in the analysis.” However, there is no detail for what kind of pulmonary function tests were carried out, and no explanation for the data analysis in the main text. Where? Is there any correlation between FGF23levels and pulmonary function?
5. The reviewer feels a bit sarcastic about the conclusion remark that “The strengths of our study includes the novelty of the findings, with significant associations with exacerbations and supplemental oxygen use”. The reviewer thinks that oxygen use should be correlated to reduced lung function. This should be put this way, with significant associations with FGF23 elevations and supplemental oxygen use.
Minor points.
1. Figure 1 does not show the correct mean value 79.5 for >2 exacerbations, as seen in the text. It looks around 50. Why does this happen?
2. In Figure 3, the right panel displays a purple shaded area with a fitting line, but no explanation. This requires explanation in the figure legend.
Author Response
We greatly appreciate the opportunity to submit a revised manuscript that was improved according to the critiques brought up by Reviewer 1.We believe that we have addressed the questions and discussed the concerns in the text of the manuscript and in this letter. With these changes, we hope that the manuscript will be acceptable for publication.
With warmest regards,
Stefanie Krick
Reviewer 1:
FGF23 plays an essential role in maintaining serum phosphate homeostasis. It has been shown that plasma concentrations of FGF23 are elevated and strongly associated with the risk of chronic kidney disease CKD progression, cardiovascular events and mortality. FGF23 is also elevated in Chronic Obstructive Pulmonary Disease (COPD). In this manuscript, the authors investigated the association between plasma FGF23 levels and AECOPD (Acute exacerbation of COPD). They found that FGF23 levels were higher in frequent exacerbations. Although the sample number is very small and the demographic is an almost exclusively male patients, the results sound significant and could support the possibility that plasma FGF23 levels may serve as a potential proinflammatory mediator.
The reviewer has a few questions and some comments.
1. According to the reference (PMID: 29748308), Individuals with mild-to-moderate COPD had higher FGF23 plasma levels when compared to patients without COPD (59±12 versus 42±10 pg/ml, P=0.004; Fig. 1a). However, FGF23 was not significantly increased in severe COPD compared to controls in this study population (51 ± 18 versus 42±10 pg/ml, P=0.26). What is the difference between severe COPD and AECOPD? This is not very clear and needs some clarification.
1. Response:We apologize that we did not make this very clear. We added clarifications in the revised discussion(page 5, line 23-28). In our previous report (reference PMID: 29748308), we analyzed a different cohort – patients with mild-to-moderate COPD, which also showed a proinflammatory phenotype (elevated systemic IL-6 levels). The cohort described here, did not show increased systemic inflammation and we presume this to be one of the reasons, why we could not detect any significant differences in FGF23 plasma levels. Patients with acute COPD exacerbations (AECOPD) though are a subgroup and tend to have increased chronic systemic and airway inflammation in addition to other complications such as increased pulmonary vascular remodeling, which are all factors that could contribute to elevated FGF23 levels. These are all future directions that need to be explored.
2. Are there any available to look at direct associations between FGF23 levels and COPD exacerbations in individuals who had the exacerbation events only after the initiation of the study? This study could have been better and much more informative, if they measured FGF levels both at the beginning and end of the study.
2. Response:We fully agree with the Reviewer; unfortunately, this exploratory study was planned and approved by the IRB to only take blood samples at one time point and the prospective follow up was done through medical chart review. The frequent COPD exacerbators, who had an increase in FGF23 levels, had events before and after the blood collection. Due to the small sample size, the number of patients with only exacerbations after the initiation of the study was negligible.
3. How does individual FGF23 levels fluctuate with time?
3. Response:The Reviewer raised a very interesting and important question and we are currently collecting FGF23 levels in individuals with COPD to address this question, since this has not been done in this patient cohort. Isakova et al. addressed this question in patients with chronic kidney disease (CKD) in the CRIC cohort and could show that circulating FGF23 levels are stable over time in the majority of CKD patients, but serial measurements can identify subpopulations with rising levels and those correlate with exceptionally high risk of death (Isakova et al., reference 20). We have included this point in our revised discussion(page 6, lines 13-16).
4. In the method section, the reviewer found the statement of “Pulmonary function tests performed … were included in the analysis.” However, there is no detail for what kind of pulmonary function tests were carried out, and no explanation for the data analysis in the main text. Where? Is there any correlation between FGF23levels and pulmonary function?
4. Response:We apologize that this was not made clearer. We added in the methods that spirometry was performed assessing FEV1 and FVC and added a reference how this was standardized. There was no correlation between FGF23 levels and spirometric values, which we have shown previously (reference 23). We revised themethodsas per Reviewer’s suggestion (page 6, line 33-34).
5. The reviewer feels a bit sarcastic about the conclusion remark that “The strengths of our study includes the novelty of the findings, with significant associations with exacerbations and supplemental oxygen use”. The reviewer thinks that oxygen use should be correlated to reduced lung function. This should be put this way, with significant associations with FGF23 elevations and supplemental oxygen use.
5. Response:We apologize and revised our discussion accordingly. We did not have access to data including complete pulmonary function tests/DLCO and oxygen saturation during a 6 minute walk test, which would show their qualification for home oxygen. Therefore, we did not find any correlation with spirometric data in this cohort.
Minor points.
1. Figure 1 does not show the correct mean value 79.5 for >2 exacerbations, as seen in the text. It looks around 50. Why does this happen?
1. Response:We apologize; by using the whisker plot, the means were not shown. We revised the manuscript accordingly to show the means and standard deviation in the revised Fig. 1.
2. In Figure 3, the right panel displays a purple shaded area with a fitting line, but no explanation. This requires explanation in the figure legend
2. Response:The purple shaded area represents the 95% confidence interval of the fitted values, represented as the line. This explanation is now included in the revised results(page 4, lines 25-26).
Reviewer 2 Report
The manuscript “Biofilm formation in Acinetobacter baumannii: genotype-phenotype correlation”, describes a cross-sectional study with individuals with COPD, and their plasma samples were analyzed for intact FGF23 levels, which is associated with systemic inflammation and altered metabolism. The authors concluded that FGF23 was consider as an exacerbator phenotype of COPD. The study has clinically important results, and it is innovative, presenting also limitations which is always important to focus.
Yet, the manuscript has still some points, that need to be adjusted before publication, namely a correction of some points in the “Results” section and a profounder discussion.
Results:
- Page 3, Line 2-4: please explain why choose this classification: “* Non frequent exacerbators are defined as patients with less than 2 or no exacerbations in 12 months prior to enrollment; # Frequent exacerbators are defined as patients with 2 or more moderate and severe exacerbations in 12 months prior to enrollment.” (reference?);
- The manuscript should present the normal values for FGF23 before discussing what is high and low;
- The manuscript refers the analysis of cardiovascular diseases. What about hepatic and immune disorders, highly related to inflammatory conditions? What it verified? If not, why?
- Beta-blockers seem to have a little bit similar results to “no-meds”. How can this be possibly explained? Please, discuss it;
Discussion:
- Please discuss the fact that most part of the cohort presented an obesity state, which may also severely contribute to inflammation conditions;
- Page 5, Line 26: “in vitro and in animal models”- correct “in vitro” to italic for and “in animal models” to “in vivo” (also italic form);
- The manuscript should present more discussion related to other similar studies, even if not the same;
Material and Methods:
- Please indicate the registration reference of the trial and also the reference for the approval by the Research Ethics Committee and/or Institutional Review Board (inform consent…);
- Define CT (CT scan) before using it;
- Page 6, line 35-36: “We classified as mild those AECOPD treated in clinic, moderate if they warrant an ER visit, and severe if they warranted hospitalization.” – please justify this classification (reference?);
- Page 7, line 4: “1-way ANOVA” – correct to One-way ANOVA;
Author Response
We greatly appreciate the opportunity to submit a revised manuscript that was improved according to the critiques brought up by Reviewer 2.We believe that we have addressed the questions and discussed the concerns in the text of the manuscript and in this letter. With these changes, we hope that the manuscript will be acceptable for publication.
With warmest regards,
Stefanie Krick
Reviewer 2:
The manuscript “Biofilm formation in Acinetobacter baumannii: genotype-phenotype correlation”, describes a cross-sectional study with individuals with COPD, and their plasma samples were analyzed for intact FGF23 levels, which is associated with systemic inflammation and altered metabolism. The authors concluded that FGF23 was consider as an exacerbator phenotype of COPD. The study has clinically important results, and it is innovative, presenting also limitations which is always important to focus.
Yet, the manuscript has still some points, that need to be adjusted before publication, namely a correction of some points in the “Results” section and a profounder discussion.
Results:
- Page 3, Line 2-4: please explain why choose this classification: “* Non frequent exacerbators are defined as patients with less than 2 or no exacerbations in 12 months prior to enrollment; # Frequent exacerbators are defined as patients with 2 or more moderate and severe exacerbations in 12 months prior to enrollment.” (reference?);
Response:We apologize that we did not make this clearer. We defined the “frequent exacerbator phenotype on page 2, line 26, 27: We considered frequent exacerbators subjects with 2 or more moderate or severe events in the 12 months (Le Rouzic et al.).We revised the manuscript and also included the original publication defining this term (Hurst et al.).
- The manuscript should present the normal values for FGF23 before discussing what is high and low;
Response:We agree with the Reviewer that this is a very important point. FGF23 levels depend strongly on phosphate intake and fractional excretion of phosphate and therefore differ between populations due to differences in diet. Therefore, there is not a universal “normal value”. In CKD studies, FGF23 levels of less than 30 pg/ml were described as normal (reference Wolf et al.). In our previous study, individuals without lung disease with a similar background and diet had average FGF23 plasma levels of 42 pg/ml).
This was now included in the revised introduction(page 2, lines 11-15).
- The manuscript refers the analysis of cardiovascular diseases. What about hepatic and immune disorders, highly related to inflammatory conditions? What it verified? If not, why?
Response:We agree with the Reviewerthat hepatic and immune disorders might be other diseases that could confound our analysis. Unfortunately, we did not have any information regarding these comorbidities. Previous reviewers from previous journals we had submitted the manuscript to, had asked about cardiovascular comorbidities and we were only able to retrieve these data indirectly, because we recorded cardiovascular medications for our cohort. We did not record any “liver-specific” medications or any immunosuppressants, which could point towards an underlying immune disorder. At the time when we conducted the study, the known confounding diseases included chronic kidney disease and heart disease – therefore, we excluded patients with renal disease.
We included this aspect now in the revised discussion (page 5, lines 7-8).
- Beta-blockers seem to have a little bit similar results to “no-meds”. How can this be possibly explained? Please, discuss it;
Response:Recently, there has been shown some evidence that beta-blockers might benefit COPD patients with frequent exacerbations. That might explain that some of our frequent exacerbators are on beta blockers, but since we were not involved in patient care decisions in this cohort, these are just assumptions. This is an observation that will be important to follow up and see whether beta blocker use has an effect on FGF23 levels, but requires a bigger cohort and randomized controlled trials in the future.
Discussion:
- Please discuss the fact that most part of the cohort presented an obesity state, which may also severely contribute to inflammation conditions;
Response:Thank you for the valid comment. We have included this aspect in the revised discussion(page 6, line 9).
- Page 5, Line 26: “in vitro and in animal models”- correct “in vitro” to italic for and “in animal models” to “in vivo” (also italic form);
Response:The discussion was edited according to the Reviewer’s suggestion (now line 38).
- The manuscript should present more discussion related to other similar studies, even if not the same;
Response:The discussion was revised accordingly.
Material and Methods:
- Please indicate the registration reference of the trial and also the reference for the approval by the Research Ethics Committee and/or Institutional Review Board (inform consent…);
Response:This was a retrospective and prospective study without any intervention – therefore, this was not registered as a clinical trial, but the study was approved by the local Institutional Review Board and written consent was obtained from each patient as stated on page 6, line 35-36.
- Define CT (CT scan) before using it;
Response:We added the definition in the revised materials and methods(page 6, line 30).
- Page 6, line 35-36: “We classified as mild those AECOPD treated in clinic, moderate if they warrant an ER visit, and severe if they warranted hospitalization.” – please justify this classification (reference?);
Response:Justification for this classification was added in the revised manuscript(page 7, line 3).
- Page 7, line 4: “1-way ANOVA” – correct to One-way ANOVA;
Response:This was corrected accordingly in the revised manuscript (page 7, line 17).